# Prevalence of, and risk factors for, diabetes and prediabetes in Bangladesh: Evidence from the national survey using a multilevel Poisson regression model with a robust variance

**Mohammad Bellal Hossain**[1]*, **Md. Nuruzzaman Khan**[2], **John C. Oldroyd**[3], **Juwel Rana**[4,5], **Dianna J. Magliago**[6], **Enayet K. Chowdhury**[6,7], **Md Nazmul Karim**[6], **Rakibul M. Islam**[5,6]

1 Department of Population Sciences, University of Dhaka, Dhaka, Bangladesh, 2 Department of Population Sciences, Jatiya Kabi Kazi Nazrul Islam University, Mymensingh, Bangladesh, 3 School of Behavioral and Health Sciences, Australian Catholic University, Fitzroy, Victoria, Australia, 4 Department of Public Health, School of Health and Life Sciences, North South University, Dhaka, Bangladesh, 5 South Asian Institute for Social Transformation (SAIST), Dhaka, Bangladesh, 6 School of Public Health and Preventive Medicine, Monash University, Melbourne, Australia, 7 School of Public Health, Curtin University, Perth, Western Australia, Australia

* bellal@du.ac.bd

**Data Availability Statement:** The data underlying this study (BDHS 2017/18) are third-party data and

## Abstract

To estimate the age-standardized prevalence of diabetes and prediabetes and identify factors associated with these conditions at individual, household, and community levels. Data from 11952 Bangladeshi adults aged 18–95 years available from the most recent Bangladesh Demographic and Health Survey 2017–18 were used. Anthropometric measurements and fasting blood glucose samples were taken as part of the survey. Prevalence estimates of diabetes and prediabetes were age-standardized with direct standardization, and risk factors were identified using multilevel mix-effects Poisson regression models with robust variance. The overall age-standardised prevalence of diabetes was 9.2% (95%CI 8.7–9.7) (men: 8.8%, women: 9.6%), and prediabetes was 13.3% (95%CI 12.7–13.9) (men: 13.0%, women: 13.6%). Among people with diabetes, 61.5% were unaware that they had the condition. 35.2% took treatment regularly, and only 30.4% of them had controlled diabetes. Factors associated with an increased prevalence of having diabetes were increasing age, male, overweight/obesity, hypertension, being in the highest wealth quintile, and living in the Dhaka division. People currently employed and living in the Rangpur division were less likely to have diabetes than those currently not employed and living in the Barishal division. Diabetes and prediabetes affect a substantial proportion (over one-quarter) of the Bangladeshi adult population. Continuing surveillance and effective prevention and control measures, focusing on obesity reduction and hypertension management, are urgently needed.

are available from the MEASURE DHS Archive via the instructions included at the following link: http://dhsprogram.com/data/Using-Datasets-for-Analysis.cfm.

**Funding:** The author(s) received no specific funding for this work.

**Competing interests:** The authors declare that they have no competing interests.

## Introduction

Diabetes mellitus remains a significant contributor to the global burden of disease [1]. People with diabetes have an increased risk of developing several serious life-threatening micro-and macro-vascular complications resulting in higher medical care costs, reduced quality of life, and increased mortality [2]. The International Diabetes Federation (IDF) has estimated that 463 million adults live with diabetes worldwide in 2019, with a projected increase to 700 million by 2045 [3]. Seventy-nine percent of those with diabetes live in low- and middle-income countries (LMICs) [4]. It is projected that diabetes cases will increase by 74% in Southeast Asian countries in the next two decades, from 88 million in 2019 to 153 million by 2045 [4].

In Bangladesh, 8.4 million adults lived with diabetes in 2019 and projected to be almost double (15.0 million) by 2045 [4]. Studies, including a systematic review and meta-analysis, and national survey reports, showed that the prevalence of diabetes among adults had increased substantially in Bangladesh, from ~5% in 2001 to ~14% in 2017 [5–8]. Several studies were also conducted using BDHS 2011 data, showing that people with older age, overweight/ obesity, hypertension, and higher socioeconomic status (e.g., education level and wealth status) were associated with increasing likelihood of diabetes [9–12]. However, these studies are relatively old and overlooked the prevalence of, and risk factors for, prediabetes. In 2019, it was estimated that 3.8 million people had prediabetes in Bangladesh, which is a major challenge for the health system in Bangladesh when to with existing cases of diabetes [4].

Thus, it is necessary to study the age-standardized prevalence of and risk factors for diabetes and prediabetes in Bangladeshi adults using the latest Bangladesh Demographic and Health Survey (BDHS) 2017–18. To our knowledge, Kibria [9] has published an article using BDHS 2017–18 with significant flaws, including an inconsistent sample size compared to the BDHS 2017–2018 published report and inappropriate age-standardization. Besides, Kibria [9] did not estimate prediabetes and estimated the odds ratio for diabetes using logistic regression, which is not robust when the outcome is common [13, 14]. Based on these limitations, we aimed to estimate the age-standardized prevalence of diabetes and prediabetes in Bangladeshi adults aged 18 years and older using the latest BDHS. We also investigated factors associated with diabetes and prediabetes in Bangladeshi adults using a multilevel Poisson regression model with robust variance. Results are examined in detail according to the individual, household, and community-level characteristics.

## Materials and methods

### Study population and data collection

Bangladesh, a Southeast Asian country, currently has a 111 million population aged 18 years and older [15]. In 2017–18, the National Institute of Population Research and Training (NIPORT), the Ministry of Health and Family Welfare, Bangladesh, conducted the second BDHS survey of its kind that collected data on blood pressure, fasting blood glucose (FBG) biomarker measurements, and relevant information in addition to socio-demographic characteristics [6].

The BDHS is a nationally representative survey conducted using a two-stage stratified sample of households, including strata for rural and urban areas. Detailed survey sampling and the data collection procedure have been published in the BDHS survey report [6]. The primary sampling units (PSUs), each containing 120 households on average, were taken from the most recent 2011 Bangladesh census enumeration areas. In BDHS 2017–18, a total of 675 PSUs was selected with probability proportional to PSU size; however, 672 PSUs were included (three PSUs were not sampled due to flooding), of which 192 and 480 were from urban and rural

areas, respectively. In the second stage, 30 households per PSU were selected using systematic random sampling to provide statistically reliable estimates of health outcomes for the country as a whole for each of the eight divisions and urban and rural areas separately. Of the 20,160 selected households, interviews were completed in 19,457 households with an overall 96.5% household response rate [6]. Of these, one-fourth of the households (4864) were selected for the collection of biomarkers. A total of 14,704 (8013 women, 6691 men) respondents aged 18 + were available in the 4864 selected households for blood glucose measurement. However, 12,100 (6919 women, 5181 men) respondents aged 18 years and older had their blood glucose tested (82.3% response rate) (Fig 1).

**Analytic sample.**   Of the respondents who had their blood glucose tested, we excluded those for whom body mass index data were missing (n = 143) and those who were pregnant at the time of blood glucose measurement (n = 7). After exclusion, we had 11,952 respondents who had their blood glucose tested which was our analytical sample for analysing the prevalence and risk factors of diabetes. However, in the subsequent analyses of pre-diabetes prevalence and risk factors, respondents having diabetes (n = 1,174) were excluded, and the data of the remaining 10,779 respondents were analysed.

## Outcome: Diabetes and prediabetes

The primary outcomes in this study were diabetes and prediabetes calculated based on the fasting plasma glucose (FPG) level [6]. The HemoCue Glucose 201 DM system with plasma conversion was used to test a drop of capillary blood obtained from consenting eligible respondents from the middle or ring finger. The system automatically converted the fasting whole blood glucose measurements taken in the survey to FPG equivalent values. The respondents were asked not to eat or drink anything other than plain water for at least 8 hours before testing. The details of blood sample collection have been described in the BDHS survey report [6]. The World Health Organization (WHO) criteria for diabetes and prediabetes classification were used [16]. The variable diabetes included respondents with diabetes defined as FPG level greater than or equal to 7.0 mmol/L or those who reported using medication for diabetes. Respondents without diabetes were those with FPG levels less than 7.0 mmol/L and not taking any diabetes controlling medication. The variable prediabetes included respondents with prediabetes defined as FPG levels between 6.1 mmol/L and 6.9 mmol/L and not taking any diabetes controlling medication. Respondents without prediabetes were those with FPG levels less than 6.1 mmol/L and not taking diabetes controlling medication [16].

## Explanatory variables

Three different levels of explanatory variables of diabetes and prediabetes were identified through a comprehensive review of the literature [5, 9, 10, 17]. Individual-level factors included were participants' age, sex, BMI, educational level, working status, and hypertension. The BMI was categorized based on Asian cut-off as suggested by the WHO expert consultation due to the high risk of type 2 diabetes and cardiovascular disease in Asian people at lower BMIs than the existing WHO cut-off [18]. The presence of hypertension was defined as a systolic blood pressure ≥ 140 mmHg and/or a diastolic blood pressure ≥ 90 mmHg, and/or currently on treatment with antihypertensive medication [19]. The household wealth quintile (lowest to highest) was the household-level factor. It was derived from the household wealth index reported in the BDHS, which was constructed using principal component analysis of household's durable and non-durable assets (e.g., televisions, bicycles, sources of drinking water, sanitation facilities, and construction materials) [20]. Community-level factors included were the place of residence and administrative divisions of the country.

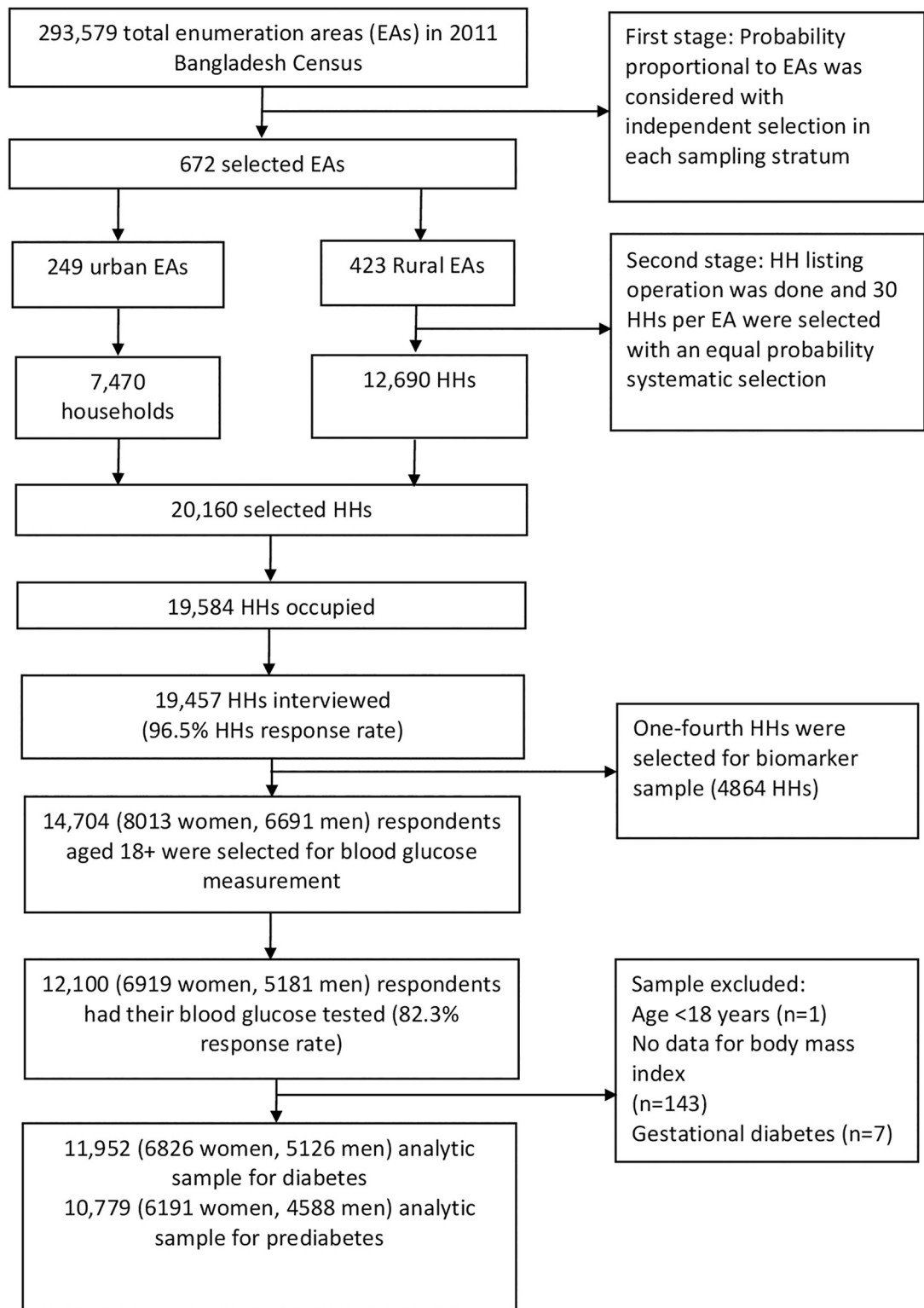

**Fig 1. Schematic representation of the sampling procedure of the Bangladesh Demographic and Health Survey, 2017–18.**

The literature review also identified family history of diabetes, mass media exposure, cigarette smoking, alcohol consumption, sleep duration, diet, and physical activity as significant behavioral and lifestyle-related individual-level risk factors for diabetes and pre-diabetes. However, this study could not include these variables in the analyses as they were not included in the original survey.

## Statistical analysis

The crude prevalence of diabetes and prediabetes were estimated, allowing for the complex survey design and survey sampling weights. To account for different age distributions between groups and over time, we age-standardized estimates to the 2011 Census population of Bangladesh using the direct method, with age categories of 18–34, 35–39, 40–44, 45–49, 50–54, 55–59, 60–64 and ≥65 years. Differences between continuous and categorical variables were tested using the Mann-Whitney and chi-square tests, respectively.

We used a multilevel Poisson regression model with a robust variance to identify factors associated with diabetes and prediabetes, and the results were presented as a prevalence ratio (PR) with a 95% confidence interval (CI). We used this model since the odds ratio estimated using logistic regression from a cross-sectional study may significantly overestimate relative risk when the outcome is common [13, 14]. Secondly, in the case of convergence failure with the log-binomial model, Poisson regression with a robust variance performs better in estimating the prevalence ratio from a cross-sectional study [21]. Furthermore, in the BDHS, individuals were nested within the household; households were nested within the PSU/cluster. Hence, our multilevel mixed-effects Poisson regression model accounts for these multiple hierarchies and dependency in data and the problem of overestimation [22].

With progressive model-building techniques, four models were run for diabetes, and separately four models for pre-diabetes, each introducing different confounding factors at the individual, household, and community levels. Model 1 was run without confounding factors to determine the cluster level variation of diabetes and pre-diabetes in Bangladesh. Model 2 and 3 were adjusted for individual, and individual plus household level factors, respectively. Model 4 was the final model that included individual, household, and community-level factors simultaneously. The Intra-Class Correlation (ICC), Akaike Information Criteria (AIC), and Bayesian Information Criteria (BIC) were used to assess model performance. All statistical tests were two-sided, and a p-value < 0·05 was considered statistically significant. The study was designed and reported following strengthening the Reporting of Observational Studies in Epidemiology (STROBE) guidelines [23]. All analyses were performed using statistical software packages Stata (version 15·10; Stata Corp LP, College Station, Texas).

## Ethical approval

This study is a secondary analysis of publicly available household survey data. Thus, we did not require any ethical approval for this study. However, institutional review boards (IRBs) at ICF and the Bangladesh Medical Research Council (BMRC) approved the survey methodology, biomarker measurements, and other survey instruments. In addition, the BDHS 2017–18 datasets are publicly available (https://dhsprogram.com/methodology/survey/survey-display-536.cfm), and we received authorization from the DHS to use the datasets.

## Results

Of 12,100 participants who provided FBG, 11,952 were included in the analysis (Fig 1). The median (IQR) age of the participants was 36 (24) years (Table 1). Of the study participants, 57.1% (6,826) were female, 26.6% (3,179) lived in urban areas, 40.1% (4,794) were overweight/

**Table 1. Characteristics of the study population aged ≥18 years in Bangladesh, BDHS 2017–18.**

| Characteristics | N (%)[a] | | | P-value[b] |
|---|---|---|---|---|
| | All (N = 11,952) | No. without diabetes (N = 10,778) | No. with diabetes (N = 1,174) | |
| **Individual level** | | | | |
| *Age in years, Median (IQR)* | 39 (24) | 35 (23) | 46 (23) | <0.001 |
| 18–34 | 5,391 (45.1) | 5,114 (47.4) | 277 (23.6) | |
| 35–39 | 1,371 (11.5) | 1,233 (11.4) | 138 (11.8) | |
| 40–44 | 1,047 (8.7) | 916 (8.5) | 131 (11.2) | |
| 45–49 | 994 (8.3) | 861 (8.0) | 133 (11.3) | |
| 50–54 | 672 (5.6) | 556 (5.2) | 115 (9.8) | |
| 55–59 | 676 (5.7) | 577 (5.4) | 100 (8.5) | |
| 60–64 | 675 (5.7) | 563 (5.2) | 112 (9.5) | |
| ≥65 | 1,126 (9.42) | 958 (8.9) | 168 (14.3) | |
| *Sex* | | | | |
| Men | 5,126 (42.9) | 4,589 (42.6) | 537 (45.8) | 0.550 |
| Women | 6,826 (57.1) | 6,189 (57.4) | 637 (54.2) | |
| *Body Mass Index (kg/m$^2$)* | | | | |
| Underweight (<18.5) | 2,067 (17.3) | 1,938 (18.0) | 129 (11.0) | <0.001 |
| Normal weight (18.5–23.0) | 5091 (42.6) | 4696 (43.6) | 395 (33.7) | |
| Overweight (23.0–27.5) | 3532 (29.6) | 3107 (28.8) | 425 (36.2) | |
| Obese (≥27.5) | 1262 (10.5) | 1037 (9.6) | 225 (19.1) | |
| *Level of education* | | | | |
| No education/preschool | 3,033 (25.4) | 2,736 (25.4) | 296 (25.2) | 0.937 |
| Primary education | 3,590 (30.0) | 3,225 (29.9) | 365 (31.1) | |
| Secondary education | 3,539 (29.6) | 3,203 (29.7) | 337 (28.7) | |
| Higher education | 1,790 (15.0) | 1,614 (15.0) | 176 (15.0) | |
| *Currently working* | | | | |
| Yes | 4,621 (38.7) | 6,695 (62.1) | 636 (54.1) | <0.001 |
| No | 7,331 (61.3) | 4,083 (37.9) | 538 (45.9) | |
| *Hypertension* | | | | |
| Yes | 3,416 (28.6) | 2,865 (26.6) | 551 (46.9) | <0.001 |
| No | 8,536 (71.4) | 7,913 (73.4) | 623 (53.1) | |
| **Household level** | | | | |
| *Wealth quintile* | | | | |
| Lowest | 2,311 (19.3) | 2,183 (20.3) | 128 (10.9) | <0.001 |
| Second | 2,354 (19.7) | 2,213 (20.5) | 141 (12.0) | |
| Middle | 2,466 (20.6) | 2,272 (21.1) | 194 (16.5) | |
| Fourth | 2,377 (19.9) | 2,111 (19.6) | 266 (22.7) | |
| Highest | 2,444 (20.5) | 1,999 (18.5) | 445 (37.9) | |
| **Community level** | | | | |
| *Place of residence* | | | | |
| Urban | 3,179 (26.6) | 2,764 (25.6) | 416 (35.4) | <0.001 |
| Rural | 8,773 (73.4) | 8,014 (74.4) | 758 (64.6) | |
| *Administrative division* | | | | |

*(Continued)*

**Table 1.** (Continued)

| Characteristics | N (%)[a] | | | P-value[b] |
|---|---|---|---|---|
| | All (N = 11,952) | No. without diabetes (N = 10,778) | No. with diabetes (N = 1,174) | |
| Barishal | 660 (5.5) | 597 (5.5) | 62 (5.3) | <0.001 |
| Chattogram | 2,053 (17.2) | 1,828 (17.0) | 224 (19.1) | |
| Dhaka | 2,767 (23.2) | 2,373 (22.0) | 394 (33.6) | |
| Khulna | 1,489 (12.5) | 1,368 (12.7) | 121 (10.3) | |
| Mymensingh | 973 (8.1) | 897 (8.3) | 77 (6.5) | |
| Rajshahi | 1,727 (14.4) | 1,590 (14.8) | 138 (11.7) | |
| Rangpur | 1,503 (12.6) | 1,419 (13.2) | 84 (7.2) | |
| Sylhet | 780 (6.5) | 706 (6.5) | 74 (6.3) | |

BDHS = Bangladesh Demographic and Health Survey; Q = quartile.

[a]All values represent absolute numbers and percentages unless otherwise stated.

[b]P-values were derived using a Mann-Whitney test or chi-square test for continuous and categorical variables, respectively.

obese and 27.4% (3,274) had hypertension. People with diabetes were significantly older than those without (p <0.001), they were significantly more likely to be overweight or obese (p <0.001), and more likely to be hypertensive (p <0.001). Furthermore, people with diabetes were more likely to live in urban areas and come from a household with higher wealth quintiles (p <0.001) (Table 1).

The crude and age-standardized prevalence of diabetes and prediabetes by individual, household, and community-level characteristics are presented in Table 2. The overall age-standardized prevalence of diabetes was 9.2% (95%CI, 8.7–9.7%) with comparable estimates for men: 8.8%, 95%CI 8.1–9.6, and women: 9.6%, 95%CI 8.9–10.3. The age-standardized diabetes prevalence was higher in urban (11.8%, 95%CI 10.9–12.7) than in rural residents (7.9%, 95% CI 7.3–8.5). Prevalence of diabetes was highest in people who were obese (18.4%, 95%CI 16.3–20.5), hypertensive (13.7%, 95%CI 12.3–15.0), in the highest wealth quintile (16.5%, 95%CI 15.9–17.9), and living in the Dhaka division (15.0%, 95%CI 13.3–16.7) compared to respective reference categories.

The overall age-standardized prevalence of prediabetes was 13.3% (95%CI, 12.7–13.9), with a similar prevalence in women (13.6%, 95%CI 12.8–14.5) and in men (13.0%, 95%CI 12.1–13.9). The age-standardized prevalence estimates of prediabetes were higher in people who were obese (18.5%, 95%CI 16.3–20.7), in the highest wealth quintile (18.6%, 95%CI 17.1–20.1), and were living in the Dhaka division (22.8%, 95%CI 20.7–24.8) (Table 2). Three out of five (61.5%, 95%CI 57.9–64.9) people living with diabetes were unaware of their condition. One-third (35.2%, 95%CI 32.0–38.5) received appropriate treatment, and only 30.4% (95%CI 26.0–35.2) of them had controlled diabetes.

Each of the four mixed-effects multilevel Poisson models was run to identify factors associated with diabetes and prediabetes. We compared intra-class correlation (ICC), Akaike's information criterion (AIC), and Bayesian information criterion (BIC) to select the best fitting model: the preferred model having the smallest ICC, AIC and BIC. According to these indicators, Model 4 (including individual, household, and community-level factors) had the best fitting model. Model 1 (crude) produced an ICC of 30.52% and 42.54% for diabetes and prediabetes, respectively (Table 3). This result indicates the degree of the variance seen across clusters without taking other factors into account. However, the ICC was reduced to 12.42% for diabetes and 14.89% for prediabetes once individual, household, and community-level factors were included in the final model.

**Table 2. Crude and age-standardized prevalence of diabetes and pre-diabetes in adults aged ≥18 years in Bangladeshi population, BDHS 2017–18.**

| | Diabetes, % (95% CI) | | Pre-diabetes, % (95% CI) | |
|---|---|---|---|---|
| | Crude prevalence | Standardized prevalence | Crude prevalence | Standardized prevalence |
| Overall | 9.8 (9.1–10.6) | 9.2 (8.7–9.7) | 14.2 (13.3–15.1) | 13.3 (12.7–13.9) |
| **Individual level** | | | | |
| *Age (years)* | | | | |
| 18–34 | 5.1 (4.4–5.9) | 5.1 (4.53–5.70) | 12.7 (11.5–14.0) | 11.5 (10.6–12.4) |
| 35–39 | 10.0 (8.4–12.0) | 10.0 (8.4–11.5) | 15.6 (13.5–17.9) | 15.2 (13.3–17.1) |
| 40–44 | 12.5 (10.4–15.1) | 11.8 (9.8–13.7) | 15.6 (13.1–18.5) | 14.9 (12.7–17.1) |
| 45–49 | 13.4 (11.2–15.9) | 12.9 (10.8–14.9) | 15.7 (13.4–18.3) | 15.4 (13.2–17.6) |
| 50–54 | 17.2 (14.2–20.7) | 16.4 (13.6–19.2) | 14.3 (11.6–17.6) | 14.7 (12.1–17.4) |
| 55–59 | 14.8 (12.0–18.0) | 15.9 (13.2–18.6) | 15.7 (12.9–18.9) | 15.3 (12.6–18.0) |
| 60–64 | 16.5 (13.7–19.8) | 15.3 (12.6–18.0) | 15.7 (12.8–19.2) | 15.4 (12.7–18.1) |
| ≥65 | 14.9 (12.6–17.5) | 14.8 (12.8–16.9) | 14.9 (12.5–17.7) | 14.7 (12.7–16.8) |
| *Sex* | | | | |
| Men | 10.5 (9.5–11.5) | 8.8 (8.1–9.6) | 14.0 (12.8–15.2) | 13.0 (12.1–13.9) |
| Women | 9.3 (8.5–10.2) | 9.6 (8.9–10.3) | 14.4 (13.3–15.5) | 13.6 (12.8–14.5) |
| *Body Mass Index (kg/m²)* | | | | |
| Underweight (<18.5) | 6.2 (5.0–7.6) | 5.3 (4.3–6.3) | 12.5 (10.8–14.4) | 11.9 (10.4–13.3) |
| Normal weight (18.5–23.0) | 7.8 (6.9–8.7) | 7.3 (6.6–8.0) | 12.9 (11.7–14.1) | 12.2 (11.3–13.1) |
| Overweight (23.0–27.5) | 12.0 (10.8–13.4) | 11.6 (10.5–12.6) | 15.3 (13.8–16.8) | 14.1 (12.9–15.2) |
| Obese (≥27.5) | 17.8 (15.6–20.3) | 18.4 (16.3–20.5) | 19.4 (16.9–22.0) | 18.5 (16.3–20.7) |
| *Level of education* | | | | |
| No education, preschool | 9.7 (8.6–11.1) | 6.9 (8.4–10.5) | 14.8 (13.2–16.6) | 13.0 (11.3–14.8) |
| Primary | 10.2 (9.0–11.5) | 9.3 (8.8–10.7) | 13.4 (12.1–14.8) | 12.7 (11.6–13.8) |
| Secondary | 9.5 (8.4–10.7) | 11.7 (8.6–10.5) | 14.3 (12.8–15.8) | 13.6 (12.4–14.8) |
| Higher education | 9.8 (8.3–11.6) | 13.1 (8.6–11.2) | 14.4 (12.5–16.5) | 15.7 (13.9–17.5) |
| *Currently working* | | | | |
| Yes | 8.7 (7.9–9.5) | 8.1 (7.5–8.7) | 13.5 (12.4–14.5) | 12.5 (11.8–13.3) |
| No | 11.6 (10.5–12.9) | 11.8 (10.9–12.8) | 15.3 (14.0–16.8) | 14.6 (13.6–15.7) |
| *Hypertension* | | | | |
| Yes | 16.4 (15.0–18.0) | 13.7 (12.3–15.0) | 14.9 (13.5–16.5) | 13.4 (12.0–14.8) |
| No | 7.3 (6.6–8.1) | 7.4 (6.8–8.0) | 13.9 (12.9–15.0) | 13.2 (12.5–13.9) |
| **Household level** | | | | |
| *Wealth quintile* | | | | |
| Lowest | 5.5 (4.4–6.9) | 5.3 (4.4–6.2) | 11.8 (10.1–13.8) | 11.5 (10.2–12.8) |
| Second | 6.0 (5.0–7.2) | 5.6 (4.7–6.6) | 10.3 (8.8–11.9) | 10.0 (8.8–11.2) |
| Middle | 7.8 (6.7–9.2) | 7.6 (6.5–8.6) | 11.7 (10.2–13.6) | 11.2 (9.8–12.3) |
| Fourth | 11.2 (9.7–12.9) | 10.4 (9.1–11.6) | 16.2 (14.4–18.2) | 14.6 (13.2–16.1) |
| Highest | 18.2 (16.5–20.1) | 16.5 (15.9–17.9) | 20.6 (18.7–22.7) | 18.6 (17.1–20.1) |
| **Community level** | | | | |
| *Place of residence* | | | | |
| Urban | 13.1 (11.8–14.5) | 11.8 (10.9–12.7) | 17.7 (16.0–19.6) | 15.1 (14.0–16.2) |
| Rural | 8.6 (7.8–9.6) | 7.9 (7.3–8.5) | 12.9 (11.8–14.0) | 12.3 (11.6–13.0) |
| *Administrative division* | | | | |
| Barishal | 9.4 (7.5–11.8) | 9.2 (7.7–10.8) | 16.3 (13.6–19.5) | 16.6 (14.5–18.7) |
| Chattogram | 10.9 (9.0–13.2) | 10.9 (9.4–12.3) | 14.6 (12.7–16.8) | 14.4 (12.7–16.1) |
| Dhaka | 14.2 (12.3–16.5) | 15.0 (13.3–16.7) | 22.0 (19.3–25.1) | 22.8 (20.7–24.8) |
| Khulna | 8.2 (6.8–9.7) | 8.0 (6.8–9.2) | 10.6 (8.8–12.9) | 10.0 (8.6–11.5) |

*(Continued)*

**Table 2.** (Continued)

| | Diabetes, % (95% CI) | | Pre-diabetes, % (95% CI) | |
|---|---|---|---|---|
| | Crude prevalence | Standardized prevalence | Crude prevalence | Standardized prevalence |
| Mymensingh | 7.9 (6.3–9.9) | 7.8 (6.3–9.2) | 12.2 (10.5–14.2) | 12.3 (10.5–14.0) |
| Rajshahi | 8.0 (6.4–9.8) | 8.2 (6.9–9.5) | 9.7 (7.8–11.9) | 9.3 (7.9–10.7) |
| Rangpur | 5.6 (4.4–7.0) | 5.6 (4.5–6.7) | 9.0 (7.3–11.1) | 9.1 (7.7–10.6) |
| Sylhet | 9.4 (7.3–12.2) | 9.6 (8.2–11.1) | 12.6 (10.5–15.0) | 12.6 (10.8–14.3) |

BDHS = Bangladesh Demographic and Health Survey; CI = confidence interval.

Data are % (95% CI). All CIs were logit transformed. Standardized to the 2011 Bangladesh Census population by age for the total population.

Diabetes is defined as a fasting blood glucose level ≥7.0 mmol/L or self-reported diabetes medication use. Prediabetes is defined as a fasting blood glucose level from 6.1 mmol/L to 6.9 mmol/L, without medication.

The results for model 4 are shown in Table 4, while the results for all other models are shown in S1 Table. The final model showed that diabetes was associated with age, sex, BMI, employment status, hypertension, wealth quintile, and administrative division of the country but not with the place of residence (urban /rural) or level of education. Compared with individuals aged 18 to 34 years, individuals aged 40 to 49 years were over two times more likely to have diabetes, while individuals aged ≥50 years were about three times more likely to have diabetes (Table 4). Men were more likely to have diabetes (PR 1.17, 95%CI 1.01–1.36) than women. Diabetes was significantly positively associated with being overweight (PR 1.23, 95% CI 1.06–1.43) or obese (PR 1.45, 95%CI 1.21–1.75) compared with normal weight, being hypertensive (PR 1.47, 95%CI 1.30–1.68) compared with normotensive, belonging to either of the fourth (PR 1.60, 95%CI 1.23–2.09); or the highest wealth quintile (PR 2.21, 95%CI 1.70–2.86), compared with the lowest quintile, and living in the Dhaka division (PR 1.32, 95%CI 1.02–1.71) compared with the Barishal division. Individuals currently employed (PR 0.81, 95%

**Table 3. Results from the random intercept model (measure of variation) at cluster/community level for diabetes and prediabetes in adults aged ≥18 years in Bangladeshi population, BDHS 2017–18.**

| Random effects (measure of variation for diabetes)[a] | Diabetes[±] | | | | Prediabetes[±] | | | |
|---|---|---|---|---|---|---|---|---|
| | Model 1 | Model 2 | Model 3 | Model 4 | Model 1 | Model 2 | Model 3 | Model 4 |
| Cluster-level variance (SE)[b] | 0.31 (0.04) | 0.23 (0.04) | 0.16 (0.04) | 0.13 (0.04) | 0.28 (0.03) | 0.26 (0.03) | 0.21 (0.03) | 0.15 (0.03) |
| Intra-class correlation (ICC, %) | 30.52% | 22.57% | 15.53% | 12.47% | 42.54% | 25.27% | 20.54% | 14.89% |
| Explained variance (PCV)(%) | Reference | 25.81% | 48.38% | 61.29% | Reference | 7.14% | 25.00% | 46.42% |
| **Model summary** | | | | | | | | |
| AIC | 7695.98 | 7264.89 | 7176.57 | 7156.65 | 9857.34 | 9849.54 | 9800.16 | 9733.91 |
| BIC | 7710.79 | 7398.19 | 7339.49 | 7378.80 | 9872.15 | 99.82.83 | 9963.07 | 9956.06 |

[±] Models are based on multilevel mixed-effects Poisson regression.

[a]We assume that the within cluster-level random effects are equal for the 'moderate' and 'highest" levels; therefore, only between cluster-level variance estimates are reported.

[b]Significance of random effects evaluated by comparing the model with a similar one in which random effects were constrained to zero.

Model 1 is the null model, a baseline model without any determinant variable

Model 2 is adjusted for individual level factors

Model 3 is adjusted for individual level plus household level factors

Model 4 is adjusted for individual level plus household level plus community level factors

The results of all models are presented in S1 Table and the results of the final model are presented in Table 4.

AIC = Akaike's Information Criterion; BDHS = Bangladesh Demographic and Health Survey; BIC = Bayesian Information Criteria; SE = Standard Error.

**Table 4. Factors associated with diabetes and prediabetes in adults aged ≥18 years in Bangladeshi population, BDHS 2017–18.**

| Characteristics | Diabetes, PR (95% CI)± | Pre-diabetes, PR (95% CI)± |
|---|---|---|
| **Individual level** | | |
| *Age in years, (ref: 18–34)* | | |
| 35–39 | 1.80 (1.45–2.25) | 1.19 (1.01–1.41) |
| 40–44 | 2.37 (1.87–3.00) | 1.22 (1.01–1.48) |
| 45–49 | 2.39 (1.89–3.01) | 1.22 (1.02–1.45) |
| 50–54 | 3.16 (2.45–4.06) | 1.12 (0.90–1.41) |
| 55–59 | 2.70 (2.13–3.43) | 1.26 (1.01–1.58) |
| 60–64 | 3.11 (2.46–3.92) | 1.26 (1.00–1.59) |
| ≥65 | 2.77 (2.16–3.55) | 1.21 (0.99–1.49) |
| *Sex, (ref: women)* | 1.17 (1.01–1.36) | 0.98 (0.87–1.10) |
| *Body Mass Index (kg/m², (ref: normal weight)* | | |
| Underweight (<18.5) | 0.83 (0.67–1.03) | 0.97 (0.83–1.13) |
| Overweight (23.0–27.5) | 1.23 (1.06–1.43) | 1.07 (0.95–1.21) |
| Obese (≥27.5) | 1.45 (1.21–1.75) | 1.23 (1.05–1.44) |
| *Level of education, (ref: higher education)* | | |
| No education, preschool | 0.93 (0.74–1.18) | 1.16 (0.95–1.42) |
| Primary | 1.16 (0.95–1.42) | 1.04 (0.88–1.24) |
| Secondary | 1.08 (0.90–1.30) | 1.05 (0.89–1.23) |
| *Currently working, (ref: no)* | 0.81 (0.69–0.94) | 1.01 (0.90–1.14) |
| *Hypertension, (ref: no)* | 1.47 (1.30–1.68) | 0.98 (0.87–1.10) |
| **Household level** | | |
| *Wealth quintile, (ref: lowest)* | | |
| Second | 1.04 (0.79–1.35) | 0.82 (0.66–1.03) |
| Middle | 1.24 (0.96–1.60) | 0.94 (0.77–1.15) |
| Fourth | 1.60 (1.23–2.09) | 1.16 (0.95–1.42) |
| Highest | 2.21 (1.70–2.86) | 1.36 (1.10–1.68) |
| **Community level** | | |
| *Place of residence, (ref: rural)* | 1.02 (0.89–1.18) | 1.00 (0.88–1.14) |
| *Administrative division, (ref: Barishal)* | | |
| Chattogram | 0.98 (0.74–1.29) | 0.82 (0.66–1.03) |
| Dhaka | 1.32 (1.02–1.71) | 1.19 (0.94–1.49) |
| Khulna | 0.78 (0.60–1.01) | 0.59 (0.46–0.77) |
| Mymensingh | 0.94 (0.70–1.27) | 0.74 (0.59–0.93) |
| Rajshahi | 0.90 (0.68–1.20) | 0.57 (0.43–0.75) |
| Rangpur | 0.67 (0.50–0.91) | 0.54 (0.42–0.71) |
| Sylhet | 1.00 (0.72–1.38) | 0.73 (0.56–0.94) |

± The final model adjusted for the individual, household, and community level factors using multilevel mixed-effects Poisson regression. The results of all models are presented in S1 Table.

BDHS = Bangladesh Demographic and Health Survey; PR = prevalence ratio; CI = confidence interval.

CI 0.69–0.94) and living in the Rangpur division (PR 0.67, 95%CI 0.50–0.91) were less likely to have diabetes than being employed and living in the Barishal division.

The fully adjusted model showed that compared with the younger age group, individuals aged between 35 to 49 years were 19% to 23% more likely to have prediabetes, and individuals aged 55 to 64 years were 26% more likely to have prediabetes. Being obese (PR 1.23, 95%CI

1.05–1.44) and belonging to the highest wealth quantile (PR 1.36, 95%CI 1.10–1.68) were associated with prediabetes compared with being normal weight and being in the lowest wealth quintile. The respondents living in Khulna, Mymensingh, Rajshahi, Rangpur, and Sylhet divisions were about 25% to 45% less likely to have prediabetes than the Barishal division. Sex, level of education, working status, hypertension, and place of residence were not associated with prediabetes.

## Discussion

Diabetes and prediabetes affect a substantial proportion of the Bangladeshi population. Based on data from the latest BDHS 2017–18, over one-quarter of individuals aged 18 years and older had diabetes or prediabetes in Bangladesh, representing more than 19 million individuals in 2020. Factors associated with diabetes were age, sex, BMI, wealth quintile, employment status, hypertension, and administrative division of the country but not the place of residence (urban /rural) or education level. These findings confirm a continuing high burden of diabetes and prediabetes in Bangladesh.

We reported that the prevalence of diabetes is lower than the overall age-adjusted diabetes prevalence of 8.7% in the Southeast Asian region, estimated by the IDF in 2021 [4]. The IDF has identified that the countries with the largest numbers of adults with diabetes aged 20–79 years in 2019 in the region are China (116 million cases) and India (77 million cases) [4]. In 2021, the IDF ranked Bangladesh 8th of countries with the highest number of adults (20–79 years) with diabetes (13.1 million cases), and it is expected to be ranked 7th in 2045 [4], consistent with our estimates. In our analysis, about 1 in 10 adults (18+) had diabetes, representing an estimated over 7.9 million individuals in Bangladesh in 2020. Note that our data included the younger population compared with the IDF estimates; as such, the total number of cases is deflated due to a very low prevalence of diabetes in the younger population. Nevertheless, this large number of diabetes cases in Bangladesh indicates that it is one of the leading countries for diabetes burden in the Southeast Asian region and highlights the urgent need for policies supporting the rollout of diabetes prevention in this country.

Our study undertook an identical methodological approach (e.g., anthropometric measurements and fasting blood samples) to the 2011 study and demonstrated that the prevalence of diabetes had increased markedly in Bangladesh over seven years [5]. Similar increasing trends of diabetes have been observed in other Southeast Asian countries [24]. However, the extent to which changes in traditional diabetes risk factors can explain the increasing trends in the prevalence of diabetes in this setting needs further investigation. A higher diabetes prevalence suggests that despite greater global awareness of diabetes and interventions for improved non-communicable disease management in primary health care [25], diabetes in Bangladesh is increasing. Furthermore, it suggests that health promotion may be failing in the face of dietary and lifestyle patterns. Thus, more resources are needed to be invested in primary health care to address the prevention of diabetes in Bangladesh.

Our estimates suggest that the prevalence of prediabetes has decreased in Bangladesh in the seven years between 2011 and 2017/18 [5]. Prediabetes is important as during this stage, micro-vascular complications occur, often without people knowing they are glucose intolerant. The literature shows that up to 40.5% of individuals with prediabetes convert to diabetes during follow-up [26]. A high conversion rate of prediabetes to diabetes is indicative of the potential for an uncontrolled increase in the prevalence of diabetes. The observation that diabetes prevalence has increased but prediabetes has decreased in seven years may indicate higher than expected conversion rates due to rapidly changing environmental conditions [25]. There is a well-established relationship between increasing age and the risk of diabetes which is

consistent with our study [4, 27]. The risks are even higher among the respondents' aged ≥50 years. The most important factors leading to such increasing risks are (i) deficiency of insulin secretion developing with age and (ii) growing insulin resistance caused by a change in body composition and sarcopaenia [28, 29]. The implication is that with increasing life expectancy in Bangladesh (current life expectancy at birth 72.3 years) [30], the increasing numbers of older people will result in even more cases and a higher burden of diabetes.

Consistent with our study, the literature shows that obesity is a leading risk factor for type 2 diabetes [31]. However, the association between obesity and diabetes is complicated as obesity is also related to socioeconomic status. We observed a significantly higher prevalence of diabetes in the highest wealth quintile than the lowest wealth quintile. A possible explanation is that those in the highest wealth quintile in LMICs use disposable income to purchase western, high-energy food ('nutrition transition') and avoid physically demanding tasks as symbols of status [32]. This results in obesity which in turn is associated with diabetes. Increasing obesity in Bangladesh [33] may also be due to reduced physical activity associated with changing traditional agricultural/domestic works replaced by technology, watching television, and using the internet. Irrespective of the explanation, increasing obesity in Bangladesh suggests that diabetes will increase further with its strengthening economy. A further policy implication is that interventions for diabetes prevention in Bangladesh need to focus on obesity, particularly by reducing the consumption of unhealthy diets and increasing physical activity. This should be given priority as there is evidence that even a modest weight reduction of 5–7% in high-risk individuals result in a decline in the incidence of diabetes [34] as recent studies demonstrated that the incidence of diagnosed diabetes is stabilizing or declining in many high-income countries since 2010 [35, 36].

Our data also show that diabetes is higher in people with hypertension, which agrees with other studies [37, 38]. In addition, hypertension is exacerbated by other risk factors such as obesity, advanced age and significantly contributes to micro and macrovascular complications resulting in renal failure and cardiovascular disease [37, 39]. Pathways through which these complications may occur include insulin resistance, inflammation, and obesity [40]. The implication of the strong association between hypertension and diabetes is that efforts are needed in Bangladesh to delay or prevent comorbid hypertension in diabetes through frequent follow-up and aggressive management.

Our study found that awareness, treatment, and control of diabetes are low in Bangladesh. Renewed efforts are needed to increase awareness, treatment, and control to improve diabetes outcomes and reduce/delay complications. Receiving appropriate treatment may be partly influenced by out-of-pocket health costs, large in Bangladesh [41]. This places a financial burden on households and has the effect of preventing people from accessing care (health care is viewed as a 'luxury' not a 'necessity') or seeking alternative providers who are cheaper but untrained and cause adverse effects [42]. Such barriers need to be addressed as they may undermine diabetes prevention efforts.

The likelihood of diabetes was high among respondents of the Dhaka division and low among respondents of the Rangpur and Barishal divisions. Many factors might contribute to such differences in risks of diabetes across divisions. For instance, being the capital of Bangladesh, the Dhaka division is highly urbanised and inhabited by people with high levels of education. Consequently, sedentary lifestyles and dependency on western foods are highly prevalent among residents of Dhaka as compared to the residents of other divisions in Bangladesh. Dhaka is also commonly characterised by very poor environmental conditions, including extreme air pollution, and few parks and open spaces. Together these factors contribute to an increased risk of diabetes among residents of the Dhaka division. In contrast, the Rangpur division is mostly rural, and people are largely employed in agricultural activity. These

characteristics could be responsible for a lower risk of diabetes among residents of the Rangpur division. These regional variations in diabetes and prediabetes prevalence in Bangladesh are consistent with studies conducted in other South Asian countries, including India and Nepal [43, 44].

The strengths of the study are that it used a large, nationally representative dataset suggesting the findings have external validity. A further strength is that clinical variables, including FBG, blood pressure, body weight, and height, were measured using high-quality techniques. The WHO criteria for the classification of diabetes and prediabetes were used, while hypertension was defined using the seventh report of the Joint National Committee on Prevention, Detection, Evaluation, and Treatment of high blood pressure criteria. Our multilevel mixed-effects Poisson regression corrects the overestimation of the effects size produced by conventional logistic regression employed in cross-sectional studies and increases the precision of the findings. However, this was a cross-sectional study, which limits our ability to infer causal relationships. The oral glucose tolerance test (OGTT) or HbA1c tests are the gold standards for diagnosing type 2 diabetes. However, using these gold standards to diagnose diabetes was not possible for population-level data in the context of a resource-poor country. Behavioral factors (e.g., alcohol consumption, smoking status, sleep duration), dietary factors (e.g., type and amount of food taken), physical activity, and genetic factors (e.g., family history of diabetes) are crucial risk factors for diabetes. However, these data were not collected and could not be controlled for in the analyses.

## Conclusions

This study implies that efforts to control diabetes and prediabetes in Bangladesh need to be strengthened and optimized, investing further resources. Given that diabetes and prediabetes are preventable diseases by modifying diet and physical activities, Bangladesh needs to intensify its efforts to implement diabetes prevention. This may require the health care system changes in which non-communicable disease prevention is prioritized and household medical care payments reviewed to reduce out-of-pocket expenses. These measures will be worth the investment as they will maximize access to high-quality public health programs. Our analysis further implies that diabetes prevention should focus on reducing obesity and managing hypertension, suggesting that their management will bring the greatest benefits. Without effective preventive measures, diabetes will continue to increase in Bangladesh.

Diabetes and prediabetes affect a substantial proportion (over one-quarter) of the Bangladeshi adult population. Despite worldwide recognition of the increasing burden of diabetes in LMICs and widespread awareness of the need for prevention through lifestyle interventions, these conditions remain a significant public health burden in Bangladesh. Along with obesity and hypertension management, newer approaches to prevention are needed, which address obesogenic environments. These will include creating walkable neighborhoods, encouraging healthy food choices in schools, and workplaces, motivating physical activity and supporting active transport. These should be part of policies to address non-communicable diseases, including diabetes, in Bangladesh.

## Supporting information

**S1 Table. Factors associated with diabetes and prediabetes in adults aged ≥18 years in Bangladeshi population, BDHS 2017–18 (all models).**
(DOCX)

## Acknowledgments

The authors thank MEASURE DHS for granting access to the BDHS 2017–18 data.

## Author Contributions

**Conceptualization:** Mohammad Bellal Hossain, Md. Nuruzzaman Khan, Juwel Rana, Dianna J. Magliago, Enayet K. Chowdhury, Md Nazmul Karim, Rakibul M. Islam.

**Data curation:** Mohammad Bellal Hossain, Md. Nuruzzaman Khan, Juwel Rana, Enayet K. Chowdhury, Rakibul M. Islam.

**Formal analysis:** Md. Nuruzzaman Khan, Juwel Rana, Rakibul M. Islam.

**Methodology:** Mohammad Bellal Hossain, Md. Nuruzzaman Khan, John C. Oldroyd, Juwel Rana, Md Nazmul Karim, Rakibul M. Islam.

**Supervision:** Mohammad Bellal Hossain, John C. Oldroyd, Dianna J. Magliago, Enayet K. Chowdhury, Rakibul M. Islam.

**Writing – original draft:** Mohammad Bellal Hossain, Md. Nuruzzaman Khan, John C. Oldroyd, Rakibul M. Islam.

**Writing – review & editing:** Mohammad Bellal Hossain, Md. Nuruzzaman Khan, John C. Oldroyd, Juwel Rana, Dianna J. Magliago, Enayet K. Chowdhury, Md Nazmul Karim, Rakibul M. Islam.

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
