## [Decision Letter · Decision Letter 0]

17 Jan 2022

PGPH-D-21-00461

Prevalence of, and risk factors for, diabetes and prediabetes in Bangladesh: Evidence from the national survey using a multilevel Poisson regression model with a robust variance

Dear Dr. Hossain,

Thank you for submitting your manuscript to PLOS Global Public Health. After careful consideration, we feel that it has merit but does not fully meet PLOS Global Public Health’s publication criteria as it currently stands. Therefore, we invite you to submit a revised version of the manuscript that addresses the points raised during the review process.

We look forward to receiving your revised manuscript.

Kind regards,

Abdur Razzaque Sarker, PhD

Academic Editor

Journal Requirements:

1. Please provide separate figure files in .tif or .eps format only and ensure that all files are under our size limit of 20MB.

2. We have noticed that you have uploaded supporting information but you have not included a list of legends.  Please add a full list of legends for all supporting information files (including figures, table and data files) after the references list.

Additional Editor Comments (if provided):

Please explore more on regional differences. Was there any differences in age, obesity, socioeconomic status, employment etc. between regions? Is there any information related to life-style behavior including caffeinated drinks, smoking and tobacco consumptions related information in BDHS dataset? If yes, these should be included in your model.

Reviewers' comments:

Reviewer's Responses to Questions

**Comments to the Author**

1. Does this manuscript meet PLOS Global Public Health’s publication criteria? Is the manuscript technically sound, and do the data support the conclusions? The manuscript must describe methodologically and ethically rigorous research with conclusions that are appropriately drawn based on the data presented.

Reviewer #1: Partly

Reviewer #2: Yes

2. Has the statistical analysis been performed appropriately and rigorously?

Reviewer #1: Yes

Reviewer #2: Yes

3. Have the authors made all data underlying the findings in their manuscript fully available (please refer to the Data Availability Statement at the start of the manuscript PDF file)?

Reviewer #1: Yes

Reviewer #2: Yes

4. Is the manuscript presented in an intelligible fashion and written in standard English?

Reviewer #1: Yes

Reviewer #2: Yes

5. Review Comments to the Author

Reviewer #1: This study aims to assess the prevalence and associated factors of diabetes and prediabetes among Bangladeshi adults. This study will provide some evidence for literature review. However, some shortage should be addressed:

#

Discussion on the condition of diabetes among several divisions should have been added as you have mentioned "administrative division" as an associated factor.

#

Some explanation should be added on the selection of references to determine the risk factors of diabetes and prediabetes.

Reviewer #2: Review on “Prevalence of, and risk factors for, diabetes and prediabetes in Bangladesh: Evidence from the national survey using a multilevel Poisson regression model with a robust–Manuscript Draft”

Thank you for the opportunity to review this manuscript. This paper addresses an important topic about the assessment of prevalence of, and risk factors for, diabetes and prediabetes in Bangladesh using the national survey. However, there are several critical insufficiencies in this manuscript have to be addressed.

Minor comments

1. Authors skipped a crucial risk factor of diabetes, cigarette smoking which should be justified. In short, lifestyle behaviour (alcohol consumption, smoking status, sleep duration, diet, and physical activity level etc.) should be acknowledged. If not, then the reason must be justified.

2. Also, family history is an established risk factor of diabetes which has been clearly avoided in this paper.

3. Mass media exposure can influence the lifestyle and behavioral pattern of an individual which in turn has direct impact on diabetes. This paper didn’t address mass media exposure as an associated factor.

4. In page 9, line 258-259, it is mentioned that “individuals aged ≥50 years were about three times more likely to have diabetes (Table 4)”. 60-64 age group has the similar result. Authors need to focus on this issue and put more emphasis and describe in the discussion section.

Major comments

1. Materials and Method (Page 4, Line 124-125)

In this study, it’s not clear if you have 2 separate models for diabetes and prediabetes or not. In this case, it is no problem to categorize study subjects into diabetes and non-diabetes, but if you categorize the same subjects into prediabetes and non-prediabetes it’s not appropriate. If so, then it will combine diabetes and non-prediabetes into one single group which doesn’t make sense for your study objective.

2. It is suggested to show the trend of diabetes and prediabetes in Bangladesh to make this manuscript more informative.

Overall Comment

Authors need to justify the necessity of this particular study when there exist similar studies as it does not appear to add anything new to what is already known about diabetes and prediabetes in Bangladesh (ref: Sarker AR 2021).

6. PLOS authors have the option to publish the peer review history of their article (what does this mean?). If published, this will include your full peer review and any attached files.

**Do you want your identity to be public for this peer review?** For information about this choice, including consent withdrawal, please see our Privacy Policy.

Reviewer #1: No

Reviewer #2: No

---

## [Decision Letter · Decision Letter 1]

12 Apr 2022

PGPH-D-21-00461R1

Prevalence of, and risk factors for, diabetes and prediabetes in Bangladesh: Evidence from the national survey using a multilevel Poisson regression model with a robust variance

Dear Dr. Hossain,

Thank you for submitting your manuscript to PLOS Global Public Health. After careful consideration, we feel that it has merit but does not fully meet PLOS Global Public Health’s publication criteria as it currently stands. Therefore, we invite you to submit a revised version of the manuscript that addresses the points raised during the review process.

We look forward to receiving your revised manuscript.

Kind regards,

Abdur Razzaque Sarker

Academic Editor

Journal Requirements:

1. Your co-authors, Md. Nuruzzaman Khan (sumonrupop@gmail.com), John C Oldroyd (John.Oldroyd@acu.edu.au), Dianna J Magliago (Dianna.Magliano@monash.edu), Enayet K Chowdhury (enayet.chowdhury@monash.edu), and Md Nazmul Karim (nazmul.karim@monash.edu), have not confirmed authorship of the manuscript. We have resent them the authorship confirmation email; however please check that the above email address for them is correct and follow up personally to ensure they confirm. Please note that we cannot pass your manuscript to Production until we have received confirmations from all co-authors.

Just in case your co-authors are having difficulty confirming their authorship, you may advise them to send us an email at globalpubhealth@plos.org and we will confirm their authorship on the authors' behalf.

Additional Editor Comments (if provided):

Reviewers' comments:

Reviewer's Responses to Questions

**Comments to the Author**

1. If the authors have adequately addressed your comments raised in a previous round of review and you feel that this manuscript is now acceptable for publication, you may indicate that here to bypass the “Comments to the Author” section, enter your conflict of interest statement in the “Confidential to Editor” section, and submit your "Accept" recommendation.

Reviewer #1: All comments have been addressed

Reviewer #2: All comments have been addressed

2. Does this manuscript meet PLOS Global Public Health’s publication criteria? Is the manuscript technically sound, and do the data support the conclusions? The manuscript must describe methodologically and ethically rigorous research with conclusions that are appropriately drawn based on the data presented.

Reviewer #1: Yes

Reviewer #2: Yes

3. Has the statistical analysis been performed appropriately and rigorously?

Reviewer #1: Yes

Reviewer #2: Yes

4. Have the authors made all data underlying the findings in their manuscript fully available (please refer to the Data Availability Statement at the start of the manuscript PDF file)?

Reviewer #1: Yes

Reviewer #2: Yes

5. Is the manuscript presented in an intelligible fashion and written in standard English?

Reviewer #1: No

Reviewer #2: Yes

6. Review Comments to the Author

Reviewer #1: Minor Comments:

1. Line 127; "and the data for the" should be replaced by "and the data of the".

2. Line 143; "or those reporting" should be replaced by "or those reported".

3. Line 147; "any" should be added after "and not taking".

4. Line 149; "any" should be added after "not taking".

5. Line 357; "respondengts" spelling should be corrected.

6. Line 401; "divisions" should be changed to "division".

7. Line 404; "lifestyles" should be changed to "lifestyle".

Reviewer #2: (No Response)

7. PLOS authors have the option to publish the peer review history of their article (what does this mean?). If published, this will include your full peer review and any attached files.

**Do you want your identity to be public for this peer review?** For information about this choice, including consent withdrawal, please see our Privacy Policy.

Reviewer #1: **Yes: **Zakir Hossain

Reviewer #2: No

---

## [Editor Report · Decision Letter 2]

18 Apr 2022

Prevalence of, and risk factors for, diabetes and prediabetes in Bangladesh: Evidence from the national survey using a multilevel Poisson regression model with a robust variance

PGPH-D-21-00461R2

Dear Dr. Hossain,

We are pleased to inform you that your manuscript 'Prevalence of, and risk factors for, diabetes and prediabetes in Bangladesh: Evidence from the national survey using a multilevel Poisson regression model with a robust variance' has been provisionally accepted for publication in PLOS Global Public Health.

Best regards,

Abdur Razzaque Sarker, PhD

Academic Editor